# Myoepithelial Carcinoma Arising in a Salivary Duct Cyst of the Parotid Gland: Case Presentation

**DOI:** 10.3390/medicina59020184

**Published:** 2023-01-17

**Authors:** Michał Gontarz, Jolanta Orłowska-Heitzman, Krzysztof Gąsiorowski, Jakub Bargiel, Tomasz Marecik, Paweł Szczurowski, Jan Zapała, Grażyna Wyszyńska-Pawelec

**Affiliations:** 1Department of Cranio-Maxillofacial Surgery, Jagiellonian University Medical College, 30-688 Cracow, Poland; 2Department of Pathomorphology, Jagiellonian University Medical College, University Hospital, 30-688 Cracow, Poland

**Keywords:** salivary duct cyst, parotid gland, myoepithelial carcinoma, parotid cyst, parotid cancer, malignant parotid cyst

## Abstract

Cystic lesions observed in parotid glands are relatively rare and comprise 2–5% of all parotid primaries. A salivary duct cyst (SDC) is a true cyst representing 10% of all salivary gland cysts. The risk of malignant transformation of SDC’s epithelium is extremely rare. In the literature, only three cases of carcinoma ex SDC of the parotid gland are described. This report presents the first in the literature case of myoepithelial carcinoma (MECA) arising from a parotid SDC. A 75-year-old male patient was referred to the Department of Cranio-Maxillofacial Surgery of the Jagiellonian University in Cracow, Poland due to a cystic tumor arising from the right parotid gland. Superficial parotidectomy with facial nerve preservation was performed. Histological examination confirmed a rare case of MECA emerging from the SDC. The immunohistochemical profile of MECA ex SDC was presented. During 6 months of the follow-up, local recurrence or distant metastasis was not observed.

## 1. Introduction

The World Health Organization (WHO) 2022 Classification of Head and Neck Tumours distinguishes tumors and tumor-like lesions, which include a variety of cysts [1]. Cystic lesions observed in parotid glands are relatively rare and comprise 2–5% of parotid primaries [2]. According to Takita et al., cystic lesions in the parotid gland can be all divided into non-neoplastic cysts, benign tumors with cystic formation (Warthin tumor, cystadenoma), and malignant tumors with macrocystic change (mucoepidermoid carcinoma, acinic cell carcinoma) [2]. Non-neoplastic cysts include salivary duct cysts (SDC), lymphoepithelial cysts, HIV-associated salivary gland disease, dermoid cysts, and lymphangioma [2]. SDC is a true cyst representing 10% of all salivary gland cysts [3]. SDCs develop from the cystic dilatation of the salivary gland duct and are observed in the superficial lobe of the parotid gland and minor salivary glands [4]. A SDC is an acquired cyst, which occurs following obstruction of the salivary duct. The main causes of salivary duct obstruction are sialoliths and mucous plaques, as well as postoperative, posttraumatic, or postinflammatory stenosis [5]. However, in most cases, the specific cause of the salivary duct obstruction is unknown. A SDC is lined with double- or multi-layered columnar and/or cuboid cells: sometimes squamous or oncocytic metaplasia might be observed [3,5]. The risk of malignant transformation of the SDC’s epithelium is extremely rare. In the literature, only three cases of carcinoma ex SDC of the parotid gland are presented: undifferentiated carcinoma [6], adenocarcinoma [7], and mucoepidermoid carcinoma [8].

The current report describes the first in the literature case of myoepithelial carcinoma (MECA) arising from the parotid SDC.

## 2. Case Report

In September 2021, a 75-year-old male patient was admitted to the Department of Cranio-Maxillofacial Surgery of the Jagiellonian University in Cracow, Poland, with a ten-year history of a slowly growing, painless cystic tumor of the left parotid gland. Progression in the last period was not observed. The patient suffered from arterial hypertension, and mitral and tricuspid valve regurgitation. In 2018, the patient had a stroke of the right hemisphere of the brain, with left-sided paresis and balance disorders. Physical examination revealed a soft, movable, and fluctuating cystic tumor of the parotid gland with unchanged skin above it. In fine needle aspiration cytology (FNAC), 15 mL of brown fluid content was obtained. Any signs of peripheral facial nerve paresis were not observed. Computed tomography (CT) showed a unilateral, solitary, well-defined, thin-walled, multilocular cystic lesion with dimensions of 49 mm (ax.) × 29 mm (cor.) × 37 mm (sag.) located in the superficial lobe of the parotid gland. In the distal part of the cystic lesion, a contrast-enhancing solid tumor 9 mm in diameter was revealed (Figure 1A). Imaging of the neck, chest, and abdomen excluded regional and distant metastases.

### 2.1. Surgery

The patient was qualified for surgical treatment. A modified Blair incision was used to perform superficial parotidectomy with facial nerve preservation. During the surgery, intraoperative neuromonitoring was used to identify branches of the facial nerve (Inomed C2 Nerve Monitor). As an antibiotic prophylaxis, 1 g of Cefazolin was administrated. The superficial lobe of the parotid gland with the thin-walled cyst was excised. The tumor was located in the lumen of the cyst, which was situated in the superficial lobe, directly on the lateral aspect of the temporal, zygomatic, and buccal branches of the facial nerve (Figure 1B). The inferior pole of the parotid gland was intact. The great auricular nerve and parotid fascia were preserved. Active drainage was removed on the second day after surgery. Four days following surgery, the patient was discharged home. The healing process was uneventful with proper facial nerve function.

### 2.2. Pathology

The histopathological examination of the surgical specimen revealed a salivary gland with lobules separated by adipose tissue and preserved histological structure. Adjacent to the salivary gland a multilocular cyst with a diameter of 40 mm was found. The cyst was lined with mainly single-layered cuboid glandular cells, in some places with double-layered, and focal formation of small papillary outgrowths up to 0.1 mm in size.

One of the cyst’s cavities included a tumor with an 8-mm diameter growing from the cyst’s wall. The tumor was composed of mixed small cells, partially epithelioid, partially spindle without marked cytological atypia, with a tendency to form trabecular structures (Figure 1C). The tumor mass surrounded the glandular structures of the salivary gland with positive staining for cytokeratin 19 (CK19) (Figure 2A). The immunohistochemical profile of MECA ex SDC characterized positive staining for keratins AE1:AE3, S-100 protein, smooth muscle actin (SMA), p40, DOG1, and SOX10 (Figure 2B, Figure 3 and Figure 4A). Only limited staining for epithelial membrane antigen (EMA) was found. Immunoreactivity for chromogranin and thyroglobulin was negative (Figure 4B). The Ki-67 labeling index (LI) in the tumor was about 10% (Figure 4C).

The entire histological picture supported the diagnosis of a low-grade myoepithelial carcinoma ex-salivary duct cyst. A cluster of cancer cells was found in the cyst wall except for the well-limited remains of the tumor tissue thrust into the lumen of the cyst. Minimally infiltrative growth through the cystic wall without surrounding tissue involvement was observed. Perineural and angiolymphatic invasion was not present. MECA ex SDC was resected with adequate surgical margins (>5 mm, R0 resection). Moreover, in the surgical specimen of the superficial lobe of the parotid gland, ten small lymph nodes with features of reactive changes without metastases were also found.

### 2.3. Follow Up

After tumor board consultation, due to the radical resection and early stage of low-grade cancer pT1N0, the patient was qualified for systematic oncological follow-up in accordance with the NCCN guidelines. A follow-up ultrasound examination 3 months after the procedure was performed without signs of locoregional recurrence. During 6 months of the follow-up, locoregional recurrence was not observed and the facial nerve function was preserved. Unfortunately, in February 2022, the patient died due to acute pancreatitis with peritonitis.

## 3. Discussion

Eighty percent of SDC occurs in the parotid gland and comprises about 1–2% of parotid lesions and 10% of all salivary gland cysts [2,3]. SDC is frequently observed in patients older than 30 years old without sex predilection [2,3]. Usually, SDC is a thin-walled, unilateral, solitary cyst situated in the superficial lobe of the parotid gland. Wall thickening might occur due to hemorrhage or infection as a result of FNAC. The differential diagnosis for SDC includes cystadenoma and low-grade mucoepidermoid carcinoma with cystic formation [4]. Surgical resection is the treatment of choice. However, after FNAC, the SDC can disappear completely and the surgery should be postponed until the recurrence of the cyst.

MECAs, also known as malignant myoepitheliomas, are rare entities and are composed of neoplastic myoepithelial cells with infiltrative growth. Myoepithelial cells are observed around the acinus and intercalated ducts of the saliva [9]. MECAs account for less than 2% of all salivary gland cancers and 0.1–0.45% of all salivary gland tumors [10,11,12,13]. The most common site of occurrence is the parotid gland (76.6%) followed by the submandibular gland [11]. This was also confirmed by our previous research, in which MECA was found in three cases localized in the parotid gland, which accounted for 1.3% of epithelial salivary gland cancers and 0.37% of all epithelial salivary gland tumors [14]. Etiological factors of MECAs origin are unknown and males and females are affected equally with an average age of around 60 years [13,15,16]. MECAs can either develop de novo or can arise in a pleomorphic adenoma and myoepithelioma [13,16]. Nevertheless, to the best of our knowledge, this case report describes the first case of MECA arising in the SDC of the parotid gland.

The pathological definition of myoepithelioma remains under discussion due to the morphological diversity of the neoplastic myoepithelial cells [16]. Differential diagnoses between benign and malignant myoepitheliomas can also be problematic. In some cases, MECA might be misclassified as a benign salivary gland tumor [10]. According to Nagao et al., MECAs are distinguished by >7 mitotic figures/10 high power fields (HPFs) or Ki-67 LI > 10%. According to Kane and Bagwan, MECAs of the parotid gland might have microscopically partial or complete encapsulation [17]. Histologically, MECAs are characterized by a wide range of morphologic tumor cell types: epithelioid, clear, hyaline, spindle, and mixed [15,17]. About 60% of MECAs are low-grade cancers with quite uniform small- to intermediate-size nuclei with gently distributed chromatin and inconspicuous nucleoli [15]. The immunohistochemical profile of MECA is characterized by the expression of Vimentin, S-100 protein, and the three antibodies against keratins (CAM5.2, AE1:AE3, and 34βE12) [15,16,17,18,19]. In addition, SMA and calponin are immunoreactive in 50% and 75% of tumors, respectively [15].

According to NCCN guidelines from 2022, surgery is the MECAs treatment of choice [20]. Low-grade MECAs with a clinically negative neck do not require elective neck dissection, unlike high-grade MECAs [20]. Xiao et al.’s study presented that occult metastases after elective neck dissection are observed in 15.2% of patients with cN0 neck. This study also confirmed that high-grade MECAs are independent, significant predictors of regional nodal disease in multivariate analysis (OR = 4.42) [21]. Clinically and/or in diagnostic imaging, enlarged lymph nodes should be treated with therapeutic neck dissection with primary MECA resection [20]. Postoperative radiotherapy or radiochemotherapy should be considered in cases of high-grade T3–4a MECAs with close or positive margins, perineural/angiolymphatic invasion, and lymph node metastases found in the surgical specimen [20].

Savera et al.’s study revealed that 29% of MECAs patients die from the disease after an average of 32 months. Only high-grade MECAs are strongly correlated with poor prognosis [15,21]. On the other hand, Luo in the cohort of 290 patients, reported 5-, 10-, and 15-year overall survival (OS) of 68.9%, 53%, and 38.1%, respectively. The independent prognostic factors influencing OS were TNM stage, grade, and postoperative radiotherapy [11]. The presented case of a low-grade MECA ex SDC consisted of mixed (epithelioid and spindle) tumor cell types. Radical resection of MECA ex SDC (R0 resection) and early stage of the disease indicated only observation without adjuvant treatment. Unfortunately, the follow-up period was only 6 months (with proper facial nerve function, without locoregional recurrence) due to the patient’s death from another cause.

## 4. Conclusions

In conclusion, carcinoma ex SDC of the parotid gland is an extremely rare entity. To our knowledge, this is the first described case of the MECA arising from the parotid SDC. Surgery is the standard of care for both SDC and MECA. MECA emerging from SDC can be successfully treated with superficial parotidectomy with facial nerve preservation without recurrence. MECA, depending on the grade and stage, might require additional neck dissection as well as adjuvant treatment.

## Figures and Tables

**Figure 1 medicina-59-00184-f001:**
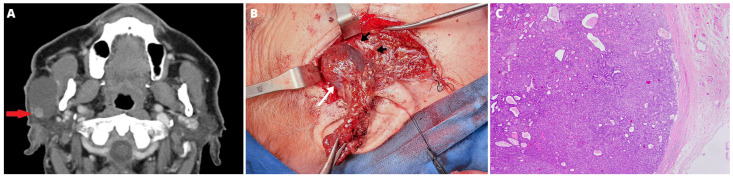
(**A**) Computed tomography (CT) scan at diagnosis showed a unilateral, solitary, well-defined, thin-walled, multilocular cystic lesion located in the superficial lobe of the right parotid gland. In the distal part of the cystic lesion, a contrast-enhancing solid tumor 9 mm in diameter was revealed (red arrow). (**B**) The thin-walled multilocular cyst (white arrow) with the superficial lobe of the parotid gland as a surgical specimen. The cyst was located directly on the lateral aspect of the temporal, zygomatic, and buccal branches of the facial nerve (black arrows). (**C**) The tumor was composed of mixed small cells, partially epithelioid, partially spindle without marked cytological atypia, with a tendency to form trabecular structures. HE, obj. magnification 200×.

**Figure 2 medicina-59-00184-f002:**
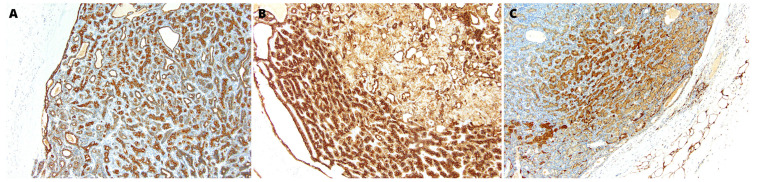
(**A**) The tumor mass surrounded the glandular structures of the salivary gland with positive staining for cytokeratin 19 (CK19). Obj. magnification 200× (**B**) The immunohistochemical profile of the tumor characterized positive staining for keratins AE1:AE3 and (**C**) S-100 protein. Obj. magnification 200×.

**Figure 3 medicina-59-00184-f003:**
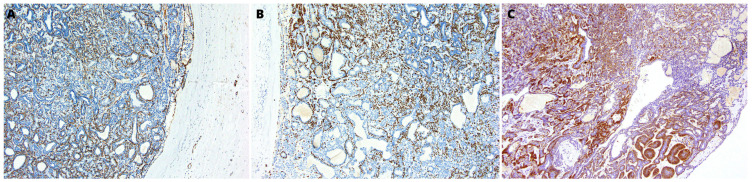
(**A**) Positive immunohistochemical staining for smooth muscle actin (SMA), (**B**) p40 and (**C**) DOG1. Obj. magnification 200×.

**Figure 4 medicina-59-00184-f004:**
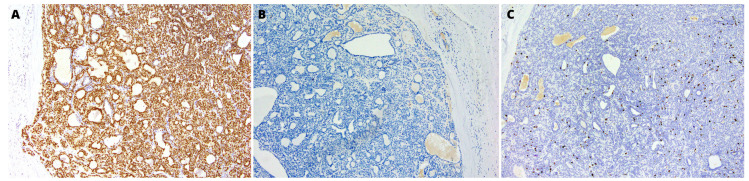
(**A**) Positive immunohistochemical staining for SOX10. (**B**) Negative immunoreactivity for chromogranin. (**C**) The Ki-67 labeling index (LI) in the tumor was about 10%. Obj. magnification 200×.

## Data Availability

Restrictions apply to the availability of these data. Data were obtained from patients treated at the Department of Cranio-Maxillofacial Surgery, Cracow, Poland, and cannot be shared, in accordance with the General Data Protection Regulation (EU) 2016/679.

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
