# Peer review of "Myoepithelial Carcinoma Arising in a Salivary Duct Cyst of the Parotid Gland: Case Presentation"

_medicina, 2023, doi:10.3390/medicina59020184_

Round 1

Reviewer 1 Report

The paper was a case review of a successful treatment for a rare disease.

- Please review for appropriate grammar. For example, the abstract is divided into two paragraphs unnecessarily, and the second paragraph starts with a number. English editing is needed in this manuscript. The authors should consider sourcing out the manuscript to a native English speaker to help with this matter

- The description of surgery is too short. Was the tumor only excised? How much was the cancer margin achieved? Deep margin? What instruments were used? Postop antibiotics?

- What does Radical surgical excision in the discussion mean?

- The conclusion is too short. It is better to add the author’s opinion on whether this disease is an easily treatable tumor by surgery (complete excision? or Radical surgical excision?).

- In Figure 1B, please identify the mass and facial nerves with arrows.

- What examinations were performed during 6-month follow-up?  

Author Response

Thank you for reviewing our manuscript. We have carefully considered all comments and revised our original manuscript accordingly. All changes in the manuscript are outlined in the red font.

Reviewer 2 Report

To try to make a deeper literature review looking for more similar cases.

A short search with google revealed some articles with the same topic even if the authors said that there is no such pathology presented yet. Everything looks fine in the paper but there are some others with the same topic:

1. Myoepithelial carcinoma of the salivary glands: a clinicopathologic study of 25 patients. A T Savera et al. Am J Surg Pathol. 2000 Jun.  

2. Myoepithelial carcinoma of the salivary glands: a clinicopathologic study of 51 cases in a tertiary cancer center. Shubhada V Kane et al. Arch Otolaryngol Head Neck Surg. 2010 Jul.;   

3. Epithelial-Myoepithelial Carcinoma of the Minor Salivary Glands: Case Series with Comprehensive Review. Diagnostics 2021, 11(11), 2124; https://doi.org/10.3390/diagnostics11112124;

  4. Parotid Myoepithelial Carcinoma in a Pediatric Patient with Multiple Recurrences: Case Report, https://www.karger.com/Article/Abstract/515783;

5. Case Rep Oncol 2021;14:989–997, https://doi.org/10.1159/000515783;

Author Response

Thank you for reviewing our manuscript. We have carefully considered all comments and revised our original manuscript accordingly. All changes in the manuscript are outlined in the red font.

The article: “Myoepithelial carcinoma of the salivary glands: a clinicopathologic study of 25 patients. A T Savera et al. Am J Surg Pathol. 2000 Jun”  has been cited before in the previous version of the manuscript.

We did not add an article: “Epithelial-Myoepithelial Carcinoma of the Minor Salivary Glands: Case Series with Comprehensive Review. Diagnostics 2021, 11(11), 2124; https://doi.org/10.3390/diagnostics11112124;” because it deals with completely different histological entity (Epithelial-Myoepithelial Carcinoma not Myoepithelial Carcinoma) located outside the parotid gland (minor salivary glands).

Round 2

Reviewer 1 Report

The revised conclusion is far from the content described thoughout this paper. The conclusion of this paper should be like this “MECA emerging from SDC can be successfully treated with superficial parotidectomy with facial nerve preservation without recurrence”

Author Response

Thank you for reviewing our manuscript. We have carefully considered all comments and revised our manuscript accordingly. All changes in the manuscript are outlined in the red font.